# An integrated system to assess marine extinctions

**Arturo Yáñez-Arenas**[1], **Miguel Nakamura**[2], **Andrew W. Trites**[3], **Héctor Reyes-Bonilla**[4], **Claudia Janetl Hernández-Camacho**[1], **Felipe Galván-Magaña**[1], **Jost Borcherding**[5], **Pablo del Monte-Luna**[1] *

**1** Instituto Politécnico Nacional, La Paz, BCS, México, **2** Centro de Investigación en Matemáticas, Guanajuato, Gto., México, **3** Institute For the Oceans and Fisheries, University of British Columbia, Vancouver BC, Canada, **4** Departamento de Biología Marina, Universidad Autónoma de Baja California Sur, La Paz, BCS, México, **5** Institute For Zoology, General Ecology & Limnology, University of Cologne, Cologne, Germany

\* pdelmontel@ipn.mx

**Data Availability Statement:** All relevant data are within the manuscript and its Supporting Information files.

## Abstract

More than 20 global marine extinctions and over 700 local extinctions have reportedly occurred during the past 500 years. However, available methods to determine how many of these species can be confidently declared true disappearances tend to be data-demanding, time-consuming, and not applicable to all taxonomic groups or scales of marine extinctions (global [G] and local [L]). We developed an integrated system to assess marine extinctions (ISAME) that can be applied to any taxonomic group at any geographic scale. We applied the ISAME method to 10 case studies to illustrate the possible ways in which the extinction status of marine species can be categorized as unverified, possibly extinct, or extinct. Of the 10 case studies we assessed, the ISAME method concludes that 6 should be categorized as unverified extinctions due to problems with species' identity and lack of reliable evidence supporting their disappearance (periwinkle—*Littoraria flammea* [G], houting—*Coregonus oxyrinchus* [G], long-spined urchin—*Diadema antillarum* [L], smalltooth sawfish—*Pristis pectinata* [L], and largetooth sawfish—*P. pristis* [L]). In contrast, ISAME classified the Guadalupe storm-petrel (*Oceanodroma macrodactyla* [G]) and the lost shark (*Carcharhinus obsolerus* [G]) as possibly extinct because the available evidence indicates that their extinction is plausible—while the largetooth sawfish [L] and Steller's sea cow (*Hydrodamalis gigas* [G]) were confirmed to be extinct. Determining whether a marine population or species is actually extinct or still extant is needed to guide conservation efforts and prevent further biodiversity losses.

## Introduction

Determining whether a species is extinct can be a difficult task, particularly for marine species [1, 2]. During the past 30 years, more than 20 global extinctions and more than 700 local extinctions (also known as extirpations or eradications) of marine species have been reported (e.g., [3–5]). However, many of these declarations are considered inconclusive or require further investigation (e.g. [6, 7]).

**Funding:** Consejo Nacional de Ciencia y Tecnología Award number: A1S19598 Recipients: Pablo del Monte Luna and Miguel Nakamura Secretaría de Educación y Posgrado, Instituto Politécnico Nacional Award number: SIP 20211495 Recipient: Pablo del Monte Luna Secretaría de Educación y Posgrado, Instituto Politécnico Nacional Award number: SIP 20200254 Recipient: Pablo del Monte Luna The funders had no role in study design, data collection and analysis, decision to publish, or preparation of the manuscript.

**Competing interests:** The authors have declared that no competing interests exist.

Although the way in which the continued existence of species is assessed—either for or against—is based on gathering documentary and empirical evidence, it does not necessarily mean that the available information has been applied in systematic, transparent, reproducible and "user-friendly" ways. Because extinctions are often used to allocate conservation resources [8], methods used to assess the status of species should be comprehensive, robust and easy-to-use to ensure that declarations of extinction can be confidently assumed to be true disappearances—or whether further investigation and data collection is required.

The methods currently used to identify extinct species can be divided into quantitative and qualitative approaches. Quantitative approaches use statistical techniques to infer extinction probabilities and extinction dates from historical sighting records, species detectability, and sampling effort. Some of these methods assume that species sightings are measured without error (e.g., [9–11]), while others incorporate the reliability and statistical uncertainty of sightings records [12–15]. For example, the IUCN Red List guidelines [16] consider existing threats to the species, sighting records, sampling effort, and the cost-benefit of declaring a species as extinct *versus* not extinct (*sensu* [8, 14, 17]).

Quantitative analyses that rely on species sightings can be difficult to apply to extinction assessments when there are few records of the species, or the only information available are last sightings [18, 19]. Some of the assessment methods rely on expert opinions to determine how reliable a record is—and frequently require information that is not easily obtained for all taxonomic groups or for different geographic scales of extinction (i.e., global and local [15, 20]). Statistical inferences associated with such methods are sensitive to how they treat the reliability of data, and whether sightings deemed to be uncertain are considered [15, 21]. Such considerations may lead to contradictory outcomes (e.g., [13]) or even inaccurate inferences—as in the case of the Aldabra banded snail, *Rhachistia aldabrae*, which reappeared in 2014 after being declared extinct with statistical confidence [20].

Recently, more sophisticated approaches based on artificial intelligence, neural networks, and machine learning have also been applied to determine the risk of extinction of threatened species under data-deficient situations [22–24]. Such methods have also been used to prioritize the conservation status of species, and allocate assessment resources [25]. Although quantitative methods provide a measure of the confidence with which species can be considered extinct, they can be difficult to resolve when the information they require is incomplete or non-existent, or require time-consuming calculations [26, 27] and advanced statistical skills.

In contrast to quantitative assessment methods, qualitative methods for assessing extinction use categorical information (e.g., empirical evidence, threatening processes, sampling intensity, and the time elapsed since a specimen was last reported) to hypothesize whether a species or population no longer exists [28, 29]. However, such qualitative assessments are not robust in the sense that they are not always easily applied to dissimilar taxonomic groups or to local extinctions. Moreover, some qualitative assessments do not consider key information, such as statistical inferences of extinction from different models, or the geographical region where a population is presumed extinct that could support or rule out declaring a species extinct. For example, one qualitative assessment approach, resulted in declaring 133 marine extinctions (21 species extinctions and the rest population extinctions [3]), while another assessment that used a wider and more updated body of documentary evidence suggested this number might be overstated by a factor of two [1]. Such differences in analytical approaches may substantially change the magnitude of extinction rates (see [18]).

Assessing extinctions in the marine realm is difficult, particularly when there is incomplete information on the species of concern, or when considerable time is required to gather the data needed to make the assessments [26, 27]. Thus, many marine species have been declared extinct or extant using assessment methods that are not reproducible or systematic (e.g., [1, 3, 30, 31]). Such methodological limitations undermine confidence in declarations of extinction.

The goal of our study was to develop a method—ISAME (Integrated System for Assessing Marine Extinctions)—that uses and evaluates all available quantitative, empirical, and documented evidence concerning the continued existence of marine species.

The ISAME method we propose is based on a comprehensive review of documented assessments of extinction (e.g., [1, 3, 9, 15, 16, 28, 29, 32]). It considers both biological and technical data associated with more than 40 declarations of marine global and local extinctions that have not been factored into previous assessments such as whether the distribution of the species is dynamic, whether the location of the purported extinction is questionable, and whether the species also inhabits other areas close to where the purported extinction occurred.

ISAME includes four hierarchical sets of binary criteria-based steps (Stages I, II, III & IV) that ultimately categorize whether a species presumed to be extinct is unverified, possibly extinct, or extinct (Fig 1). Definitions of the categories and other terms relevant to the system are given in Table 1. Stages I and II involve establishing the identity of the species and whether its status has been peer-reviewed—while Stage III involves an arithmetical score of 11 quantifiable criteria associated with distributions, sightings and other data pertaining to extinction. The sum of these 11 criteria scores culminates in assigning one of three possible statuses at Stage IV—Extinct, Possibly Extinct, and Unverified extinction.

ISAME offers advantages over other assessment methods. Firstly, it is a comprehensive tool built on over 40 documented global and local marine extinctions (excluding the information related to the 10 case studies we put through the ISAME assessment). Secondly, it incorporates both qualitative and quantitative data in a relatively simple four-staged assessment process (Stages I–IV). Thirdly, it categorizes extinct species into three distinct groups (Extinct, Possibly Extinct and Unverified) that are relevant for conservation efforts, and are applicable on global and local scales as well as at any taxonomic group. Lastly, the ISAME method has a user-friendly and intuitive interface that yields useful outputs for decision-making.

In addition to providing a detailed description of ISAME, we use data from 10 case study species to illustrate five ways in which the three categories of extinction can be derived (Fig 1). These case studies include declarations of **1)** global extinctions for the periwinkle (*Littoraria flammea*), houting (*Coregonus oxyrinchus*), Guadalupe storm-petrel (*Oceanodroma macrodactyla*), lost shark (*Carcharhinus obsolerus*), and Steller's sea cow (*Hydrodamalis gigas*); and **2)** local extinctions of long-spined urchin (*Diadema antillarum*), smalltooth sawfish (*Pristis pectinata*) from the Northwest Atlantic and Bermuda, U.K, and largetooth sawfish (*P. pristis*) from the U.S.A. waters and the Gulf of California.

## Materials and methods

We attempted to identify diverse criteria that drew on all available scientific information that one might consider to reliably decide whether a species or population is extinct. In the case of local extinctions, we incorporated additional criteria that could play a role in categorizing a species as extinct. We did so in such a way as to not affect the evaluation and categorization process in global cases. We thus settled on a set of 11 binary criteria that captured information on sampling effort (A1), statistical analysis and surveys (A2), presence over time (B1), threats (C1), proneness to extinction (C2), historical sightings (D1), extralimital distribution (D2), dynamic distribution (D3), probability of existence (E1), habitat suitability (E2), and dispersal capacity (E3). Applying the 11 criteria together yields one of two options: either there are (Stage IV-B) or there are not reasons to suspect that the species in question will be classified as extinct (Stage IV-A). Additional details of the rationale and literature supporting all criteria used in ISAME are summarized in S1 Table. Ranking of the 11 criteria reflects the relative importance that either supports or casts doubt on extinction.

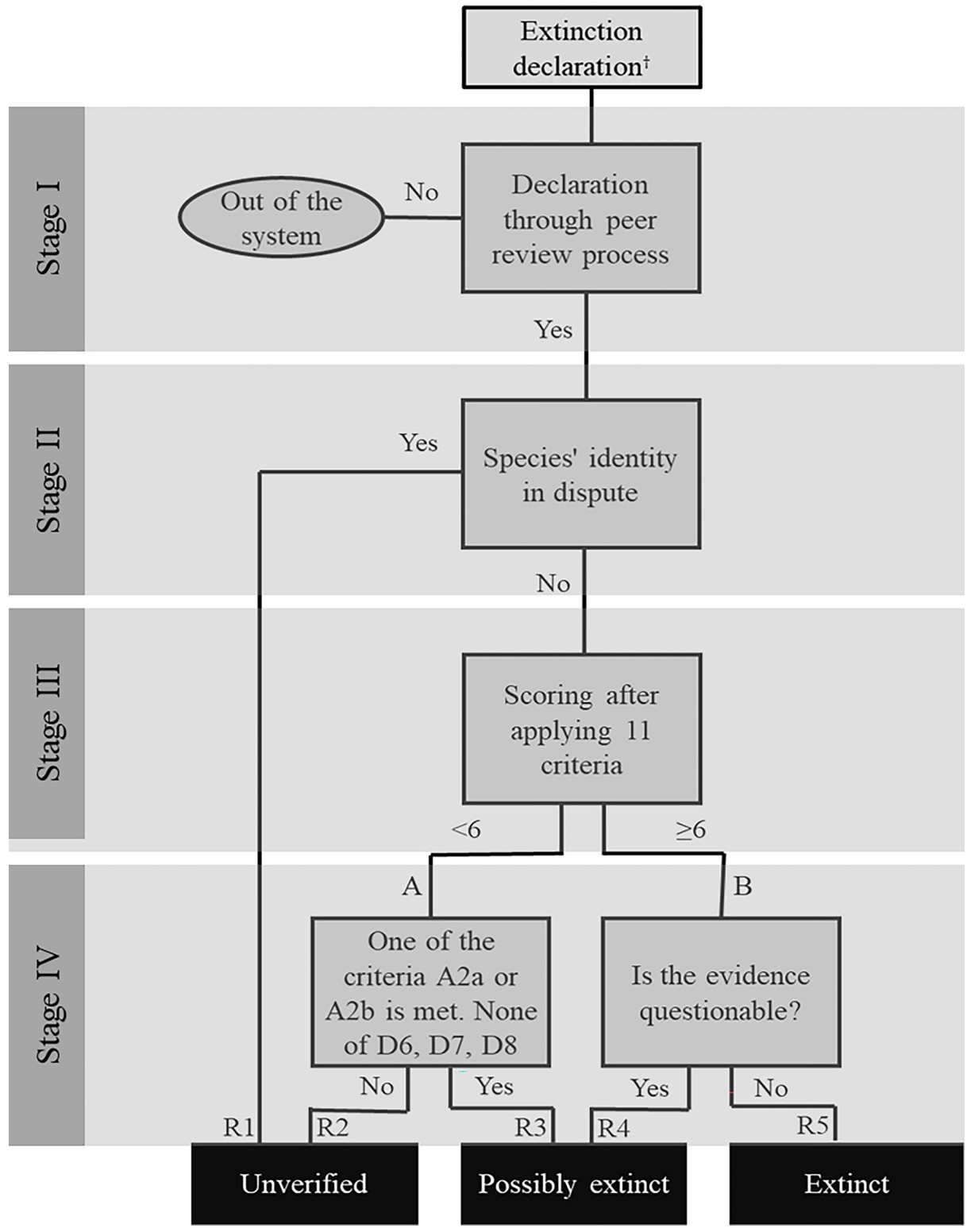

**Fig 1. Flowchart of the integrated system for assessing marine extinctions (ISAME).** Flowchart shows how one of three extinction categories (unverified, possibly extinct, and extinct) can be assigned to species presumed to be extinct based on 4 Stages of assessment (see text for full description). Alternatively, if there is a suspicion of extinction (supported by evidence) but the declaration has not yet been published, the case is treated as a candidate for further assessment.

**Table 1. Definitions of the extinction spatial scales and system categories.**

| Scale of extinction or category | Definition |
| --- | --- |
| Local extinction/ extirpation | When one or several populations within a particular habitat have completely disappeared for at least 10 years. The disappearance is not associated with migration, emigration, or with the natural dynamic distribution of the species (periodicity in the absence/presence of the species within the habitat or habitats). Let the latter argument be labeled as (M). |
| Global extinction | When there is no reasonable doubt that a species has completely disappeared worldwide |
| Unverified | When there is little or no evidence supporting that a species or population has completely disappeared locally or globally, or there is a dispute over the species' identity. In the case of local extinctions, there is supporting evidence of migration, emigration, or that the distribution of the species is naturally dynamic. |
| Possibly extinct | When it is plausible that the species has disappeared locally or globally. In local extinctions + (M). In the absence of evidence, there is a probability that the species is still extant. |
| Extinct | When there is little or no evidence to argue that a species or specie's population still exists locally or globally. In local extinctions + (M). |

Unverified may be equivalent to Data deficient or Critically endangered (depending on the case) IUCN categories; Possibly extinct may be equivalent to Critically endangered (Possibly extinct) IUCN category.

## Validating extinction criteria

The binary extinction criteria we identified give rise to $2^{11}$ = 2,048 possible ways that the 11 criteria values might occur together (no = 0, yes = 1). We reviewed all existing configurations of criteria and assigned them a presumed tag before Stage IV (presumed extinct, which means it goes to stage IV -B, or non-extinct, which means it goes to stage IV-A) based on the definition of extinction and the evidence for and/or against each configuration. For example, a species is deemed to be presumed extinct if it presents the following evidence or combination of criteria: $A1$ = 1 (there are extensive and constant sampling efforts throughout the species' range); $A2$ = 1 (there are published statistical analyses inferring the extinction date of the species); $B1$ = 1 (the species has been absent for more than 50 years); $C1$ = 1 (there is evidence of processes that have threatened the species); $C2$ = 1 (the species presents attributes correlated to its extinction); $D1$ = 0 (there are no sightings in the last 10 years); $D2$ = 0 (there are no new records outside the distribution range); $D3$ = 0 (the species' residency is not in dispute); $E1$ = 0 (there is no statistical evidence indicating a probability of the species' existence); $E2$ = 1 (the site where the species was found appears to be suitable for recolonization); and $E3$ = 0 (there is no evidence that the species is found near the site where the extinction occurred).

In contrast to ways in which the criteria lead to conclusions that a species is extinct, a species considered not extinct (due to lack of evidence or to existing uncertainty) could meet the following combination of criteria: $A1$ = 0 (there are no extensive and constant sampling efforts throughout the species' range); $A2$ = 0 (there are no published statistical analyzes inferring its extinction date); $B1$ = 1 (the species has been absent for more than 50 years); $C1$ = 1 (there is evidence of processes that have threatened the species); $C2$ = 1 (the species presents attributes correlated to its extinction), $D1$ = 0 (there are no sightings in the last 10 years); $D2$ = 0 (there are no new records outside the distribution range); $D3$ = 0 (the species' residency is not in dispute); $E1$ = 1 (there is statistical evidence indicating a probability of the species' existence); $E2$ = 0 (the site where the species was found appears to be suitable for recolonization); and $E3$ = 0 (there is no evidence that the species is found near the site where extinction occurred).

Of the 2048 combinations, we considered 2028 to be clear-cut (i.e., that reasons for arguing that a species having one of these combinations is either "presumed extinct" or "non-extinct"

would not be controversial). Within these clear-cut combinations, we tagged 114 as "Presumed Extinct" and 1914 as "Non-Extinct". The remaining 1% (i.e., 20 of the 2,048 possible combinations) were considered borderline or *contentious*, in that the resulting assigned status is not entirely clear-cut. All existing combinations can be found in "S1 File. All criteria configurations" in the supporting information. In order to practically determine extinction status without having to look up a given result in a full-fledged table of criteria configurations, we developed a numerical algorithm to easily calculate the extinction status as a function of the 11 binary responses.

## ISAME scores: Facilitating the treatment of 11 binary responses

A classification for a species meeting specific criteria is derived directly from any combination of the 11 binary responses. This can be done by consulting the S1 File containing all 2048 possible combinations. We nevertheless implemented an alternative device based on scoring in order to avoid this somewhat cumbersome table lookup. This solely amounts to an easier way of determining status based on 11 binary responses.

To develop this scoring system, we used the 2028 clear-cut combinations of criteria (omitting the 20 contentious cases) to associate each criteria group with an integer-valued weight—from which the individual weights were summered over all criteria marked as met, and then compared with a threshold for classification. More precisely, we considered three positive integers ($A$, $B$, $C$), two negative integers ($D$, $E$), and an integer threshold ($U$). Integers $D$ and $E$ are negative because answering *yes* to criteria D1, D2, D3, E1, E2, or E3 is indicative of a species *not* being extinct. The overall score ($S$), defined by adding values from each of the criteria assessed as 1, results in larger values of $S$ corresponding to greater accumulation of evidence towards extinction. For example, if a species assessment yields the binary values A1 = 1, A2 = 1, B1 = 1, C1 = 0, C2 = 0, D1 = 0, D2 = 1, D3 = 1, E1 = 1, E2 = 0, E3 = 0, the resulting score is $S = A+A+B+D+D+E$. Comparing the total score to the threshold results in a species being assessed as extinct if $S \geq U$, and as non-extinct if $S < U$.

A fixed, arbitrary set of weights $A$, $B$, $C$, $D$, $E$ and threshold $U$ implicitly induces a unique classification under the summation scheme. However, any given classification can be achieved under differing sets of constants. In other words, many sets of constants, termed *viable*, lead to perfect classifications of the 2028 clear-cut combinations. Sets of weights with the smallest magnitudes are the simplest to compute and interpret (e.g., weight 4 is simpler than 19 or 252). Conducting comprehensive case-by-case searches over viable sets of constants revealed that values $A = 4$, $B = 3$, $C = 1$, $D = -8$, $E = -1$ and $U = 6$ achieves perfect classifications for clear-cut cases, and are also simplest in terms of smallest magnitudes (that is, max($A$, $B$, $C$, $|D|$, $|E|$, $U$) is as small as possible). These numbers not only provide a simple, operational means to assign status that avoids having to consult a table spanning 2048 combinations of possible outcomes, but also represent *factor weights* that reflect the relative importance and contribution of each criterion for determining extinction status.

Our scoring algorithm uses these fixed constants (Stage III)—and relies on dichotomic choices at Stage IV to classify the species as being unverified extinct, possibly extinct, or extinct (Fig 1). Stage IV was established to more accurately classify a species into one of the three categories, and helps to classify contentious cases (see contentious cases and possible classifications—S2 Table). It is important to note that our method is not intended to establish the risk of extinction of a species, but rather to help discern whether a species is extinct or not using all available evidence.

We tested our method using from 57 cases of marine species currently declared extinct (see S2 File. Assessing cases). Of these 57 case studies, our method assessed some as unverified extinctions, others as possibly extinct, and the reminder as unquestionably extinct. The results

were discussed and assessed by a research group working in the area of conservation biology and topics related to extinctions—and then independently reviewed and questioned by experts working with the different taxonomic groups. Out of these 57 cases, we have provided in-depth details for 10 of them that we felt are representative of the different scoring criteria we propose, and the different ways in which the flow chart (Fig 1) works.

None of the 57 assessed case studies we considered were deemed to have a contentious status. In other words, the variables scored for them were not among the 20 out of 2,048 possible combinations we considered to be contentious. We conjecture that these possible 20 combinations of binary responses (see S2 Table) are unlikely to ever occur or are biologically implausible to observe in practice. Should a species of interest result in one of these 20 contentious combinations, the resulting classification should be assigned with caution. In such cases, ISAME can be improved by adding or modifying the criteria to reliably classify the new entries. There is also no conceptual limitation to adding a 12th or 13th criteria to the 11 used, or even adding a Stage V to resolve contentious cases. After applying the method to a sizable number of real cases, we consider the results obtained from the combinations to be consistent with those of the IUCN.

The ways in which the 11 binary criteria can potentially occur are provided for reference (S1 File. All criteria configurations). For practical applications, a worksheet is available in the Supporting information (S3 File. ISAME application) to assist with the bookkeeping of stages and criteria. The excel sheet can be copied or downloaded to begin on the tab labeled "Stage 1". If a criterion is met, the value of "1" must be set in its cell; if it is not met, the value of "0" is set. After reviewing whether the 11 criteria pertaining to "Stage 3" are met, the researcher will be prompted to proceed to the tabs corresponding to Stages IV-A or IV-B depending on the results. In the unlikely event that a species being evaluated belongs to the set of 20 contentious configurations, a warning message will appear to indicate that closer scrutiny is called for.

## System stages

**Stage I. Genuine declaration of extinction.**   Any declaration of extinction can be assessed using ISAME if it has undergone a formal independent peer review. Alternatively, if extinction is suspected (and supported by evidence), but the declaration has not been published, the case can still be assessed by ISAME.

**Stage II. Dispute over the species' identity.**   If the declaration meets the conditions of Stage I, the species' identity must not be under debate in the scientific literature or in an international database (e.g., Integrated Taxonomic Information System). If disputed, the case is categorized as unverified—otherwise, it passes to Stage III.

**Stage III. Scoring quantifiable criteria of extinction.**   Eleven binary (yes/no or 1/0) criteria are assessed, and assigned a criteria-specific numerical value ($-8$, $-1,0,1,3$ or $4$). We selected these specific values using an optimized systematic search procedure. Of all the possible combinations of values that might be summed for these 11 criteria, we found that 2,028 were biologically plausible, and 20 were contentious (i.e., rare or biologically implausible). The total sum of values across all criteria constitutes a *score*, which in turn determines whether the case passes to Stage IV-A (score $< 6$) or to Stage IV-B (score $\geq 6$). All 11 criteria must be evaluated. A criterion that is not met or is not applicable is assigned the value 0.

**A1.** *Sampling effort.* Sampling effort (surveys directed at the species or performed opportunistically) was constant or increased over time—and encompassed the species' distribution and detectability (i.e., relative to the species' reproductive rate, body size, behavior, etc.; Fig 2). Note that sampling effort to declare a small and cryptic species extinct should be comparatively greater than the time needed to declare a large, conspicuous species extinct. (Value +4).

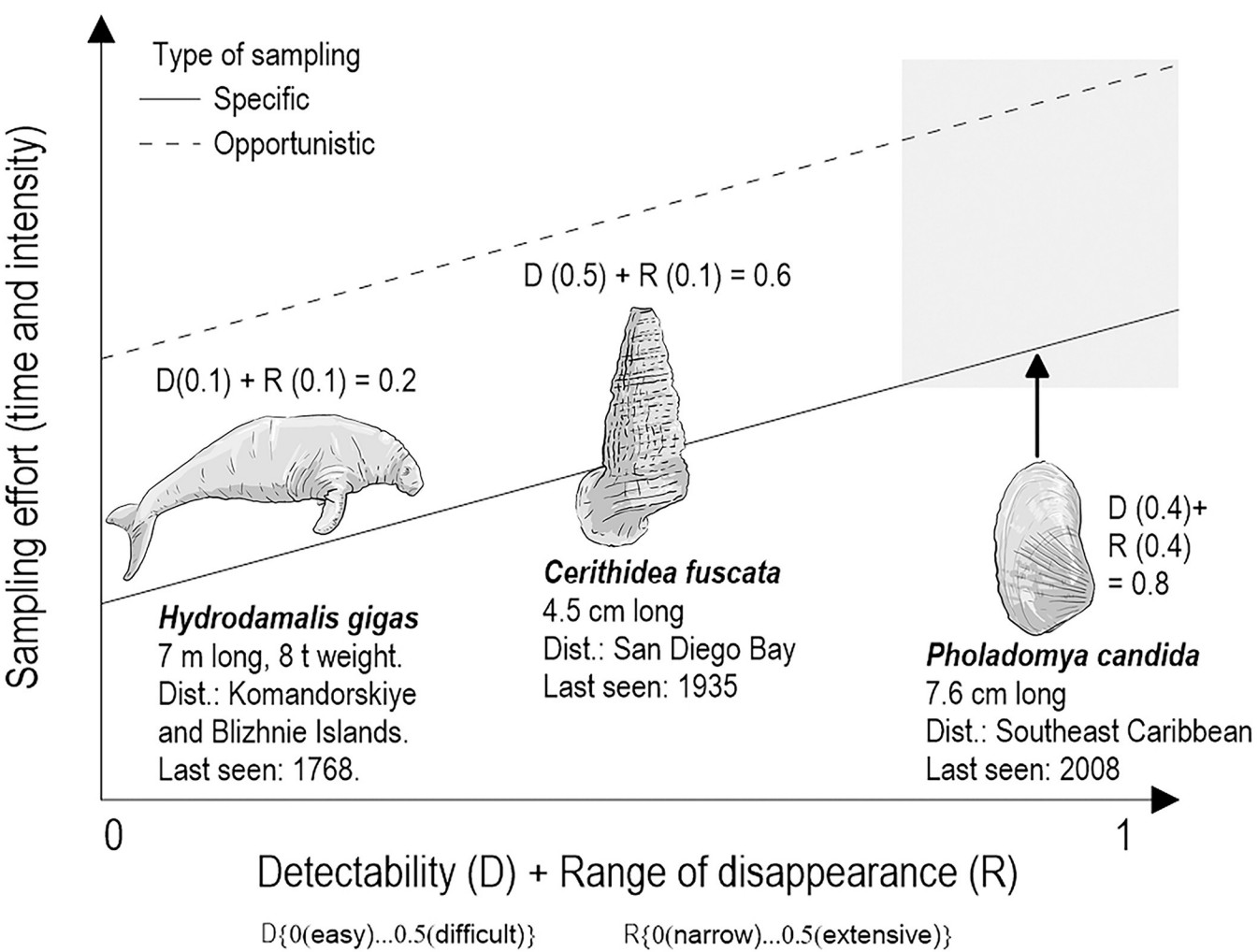

**Fig 2. Sampling effort required (specific: Solid line; opportunistic: Dashed line) to declare a species as extinct as a function of its detectability and range of disappearance.** The lower the species detectability and the more extensive its distribution range are, the more sampling effort is necessary to determine its extinction. The three species shown have been declared extinct. Two of these met the A1 criterion (see text; Steller's sea cow *Hydrodamalis gigas* and *Cerithidea fuscata*) and one did not (*Pholadomya candida*). Steller's sea cow is easily detected and has a narrow distribution range and, requires less effort to be declared extinct than *C. fuscata* (which is difficult to detect and has a narrow distribution range). The sampling effort necessary to determine the extinction of *P. candida*, which is difficult to detect and has an extensive range, should have been more intensive than that for the other two species (arrow pointing to the light gray box). The values shown are relative measurements.

**A2.** ***Statistical analysis & surveys.*** There are statistical inferences (A2a) together with extensive surveys (A2b). (Value +4).

**A2a.** ***Probability of extinction.*** Statistical inferences indicate that the species or population is extinct. The upper limits of the confidence interval of the extinction date(s) must be prior to the year when ISAME is applied.

**A2b.** ***Survey effort.*** In addition, extensive surveys (directed or opportunistic) were undertaken throughout the species range that had a high probability of detecting the species or population. Note that if A1 is affirmative, then A2b is implied to be positive.

**B1.** ***Presence over time.*** The presence of the species or population has not been recorded for >50 years. (Value +3).

**C1.** ***Threats.*** Direct and severe threats to the existence of the population or species are known (e.g., ongoing and severe overexploitation, pollution, and habitat loss). (Value +1).

**C2. *Proneness to extinction.*** The species possesses one or more attributes that are correlated with extinction proneness (e.g., inability to fly, restricted geographical distribution, late maturity, and large body size). (Value +1).

**D1. *Historical sightings.*** There are documented historical sightings in the last 10 years or over three generations. (Value −8).

**D2. *Extralimital distribution.*** There is evidence that the species occurs at sites outside its previously known distribution. However, sampling effort in the extended range has not been as intensive as in the historic range (criterion A1). Applicable only to global extinctions. (Value −8).

**D3. *Dynamic distribution.*** The distribution of the population(s) has frequently changed in time and space, or its residence in the place of purported extinction is questionable. Applicable only to local extinctions. (Value −8).

**E1. *Probability of existence.*** There is a statistical probability that the population/species still exists. The upper limit of the confidence interval of the extinction date must exceed the current year. If there is more than one estimate, the average of estimates that are 10 years before and 10 years after the current year must be calculated. If the average is prior to the current year, A2a criterion is met but not E1—and if the average is after the current year, E1 is met but not A2a. (Value −1).

**E2. *Habitat suitability.*** The site where the extinction occurred is suitable for recolonization; otherwise, there is evidence that habitat modification, ecological barriers or other threats impede recolonization. Applicable only to local extinctions. (Value −1).

**E3. *Dispersal capacity.*** The species is still present in areas near the extinction site. Additionally, the species' dispersal capacity allows individuals from locations where the species still occurs to reach the site of disappearance. Applicable only to local extinctions. (Value −1).

*Stage IV-A (Stage III score <6).* Cases entering this stage must fulfill the following conditions to be categorized as possibly extinct:

1. None of the D criteria of Stage III are met.

2. The case meets A2a or A2b.

If the case does not meet any of the previous conditions, it is categorized as unverified.

*Stage IV-B (Stage III score ≥ 6).* To be categorized as extinct at this stage, the available evidence of the records (e.g., sightings, specimens, physical remains, fossil/subfossil records, etc.) must be re-examined to ensure that it is indeed a valid species that has disappeared—or whether there is reasonable doubt. Examples of reasonable doubt include a lack of clarity regarding whether the remains on which the identification is based belong to the juvenile or adult stage of the species; the sampling conditions or specimens cannot be traced, reconstructed, or verified (e.g., purchased or found at markets). If the species or population does have issues pointing to reasonable doubt, it can be categorized as possibly extinct. An Excel spreadsheet to aid in conducting an ISAME evaluation can be found in the online Supporting information

## Results

Of the 10 presumed extinct species we chose to illustrate the five possible pathways to categorize extinctions, ISAME classified 6 as unverified, 2 as possibly extinct, and 2 as extinct (Table 2; see also Supporting information). Two of the species declared globally extinct—periwinkle and houting—were categorized as unverified because their taxonomic identities were in dispute (Route 1; Fig 1). The four others were classified by ISAME as unverified extinctions at local scales (long-spined urchin, smalltooth sawfish from the Northwest Atlantic and Bermuda, U.K., and the largetooth sawfish from the Gulf of California). None of these 6 species

**Table 2. Results of the assessment of 10 study cases representing the five possible routes of the integrated system for assessing marine extinctions.**

| Species (extinction level)† | Location‡ | SI§ | SII¶ | SIII†† | | | | | | | | | | | | SIV‡‡ | | Category§§ |
|---|---|---|---|---|---|---|---|---|---|---|---|---|---|---|---|---|---|---|
| | | | | A1 | A2 | B1 | C1 | C2 | D1 | D2 | D3 | E1 | E2 | E3 | Total | A | B | |
| *Littoraria flammea* periwinkle (G) | CN | Y | N | - | - | - | - | - | - | - | - | - | - | - | - | - | - | U |
| *Coregonus oxyrinchus* houting (G) | NS | Y | N | - | - | - | - | - | - | - | - | - | - | - | - | - | - | U |
| *Diadema antillarum* long-spined urchin (L) | C | Y | Y | 0 | 0 | 0 | 1 | 1 | −8 | 0 | 0 | 0 | −1 | −1 | −8 | N | - | U |
| *Pristis pectinata* smalltooth sawfish (L) | NWA | Y | Y | 0 | 0 | 0 | 1 | 1 | ? | 0 | −8 | −1 | ? | −1 | −8 | N | - | U |
| *P. pectinata* smalltooth sawfish (L) | B | Y | Y | 0 | 0 | 0 | 0 | 1 | 0 | 0 | −8 | 0 | 0 | 0 | −7 | N | - | U |
| *P. pristis* largetooth sawfish (L) | GC | Y | Y | 0 | 0 | ? | 0 | 1 | 0 | 0 | −8 | 0 | 0 | 0 | −7 | N | - | U |
| *Oceanodroma macrodactyla* Guadalupe storm-petrel (G) | GI | Y | Y | 0 | 0 | 3 | 1 | 1 | 0 | 0 | 0 | 0 | 0 | 0 | 5 | Y | - | PE |
| *Carcharhinus obsolerus* lost shark (G) | WCP | Y | Y | 4 | 0 | 3 | 0 | 0 | 0 | 0 | 0 | 0 | 0 | 0 | 7 | - | N | PE |
| *P. pristis* largetooth sawfish (L) | US | Y | Y | 4 | 4 | 3 | 1 | 1 | 0 | 0 | 0 | 0 | −1 | 0 | 12 | - | Y | E |
| *Hydrodamalis gigas* Steller's sea cow (G) | NWP | Y | Y | 4 | 4 | 3 | 1 | 1 | 0 | 0 | 0 | 0 | 0 | 0 | 13 | - | Y | E |

†Extinction level: local (L), global (G)

‡Locations: China (CN), North Sea (NS), Caribbean (C), Northwest Atlantic (NWA), Bermuda (B), Gulf of California (GC), Guadalupe Island (GI), Western Central Pacific (WCP), U.S. waters (US), Northwest Pacific (NWP)

§Stage I. Genuine extinction declaration. Y (yes) means that a case met the requirements of a specific stage (it proceeded to the next stage). N (no) means that it did not meet these requirements (direct category assignment)

¶Stage II. Dispute over the species' identity

††Stage III. Arithmetic score. A, B, C, D, and E: criteria codes. A question mark (?) means that the criterion information is unknown or questionable. Total: Summation of A1 –E3 that defines whether an extinction declaration goes to stage IV-A or to stage IV-B (<6 goes to stage IV-A and ≥6 goes to stage IV-B)

‡‡Stage IV-A (<6) and B (≥6)

§§Categories: unverified (U), possibly extinct (PE), extinct (E).

could be classified as extinct. They all lacked sufficient survey effort and were last seen <50 years ago (criteria A1, A2, and B1), and they all received low scores for the remaining criteria (<6; criteria D1, D3, E1, E2, and E3) (Route 2; Fig 1).

Another two species declared globally extinct were categorized by ISAME as possibly extinct. These included the Guadalupe storm petrel and the lost shark. The storm-petrel scored <6 in Stage III and met all requirements of Stage IV-A (Route 3; Fig 1), while the lost shark scored ≥ 6 in Stage III, but did not meet the requirements of Stage IV-B (Route 4; Fig 1). Finally, ISAME classified the largetooth sawfish as locally extinct and Steller's sea cow as extinct. Both study cases met most of the Stage III criteria, and the supporting evidence indicates their total disappearance beyond any reasonable doubt (Route 5; Fig 1).

## Case studies

**Route 1: Extinction unverified.** Dispute over the species' identity—the periwinkle and the houting.

The periwinkle and the houting are both considered globally extinct by the IUCN [33]. The periwinkle is distributed in China and was initially considered a rare species according to museum records [34]. However, it was later declared a strong candidate for modern marine extinction after intensive sampling failed to find it [1, 3, 30, 33]. Nevertheless, after 150 years of absence, the species was found north of Shanghai and in the Yangtze River deltas (which automatically ruled out its extinction), and genetic evidence further questioned whether *L. flammea* could actually be *Littoraria melanostoma* [35].

The houting was pronounced extinct in 2005 [36], where it was noted that *C. oxyrinchus* is a separate species from other houtings inhabiting the North and Baltic Seas. Their re-description was based on 14 museum specimens whose number of gill rakers differed from that of the reintroduced North Sea houting, which they identified as *Coregonus maraena*. However, several authors [37–40] have consistently reported the successful reintroduction of North Sea houting, referring to it instead as *C. oxyrinchus*. This discourse over the species' identity means that its extinction cannot be considered conclusive.

The taxonomic status of the houting proposed in 2005 [36] is solely based on morphometry, with no consideration of genetic evidence or ontogenetic variation. This lack of representativeness is relevant because the taxonomy of coregonids is still recognized as chaotic, and the morphology of the species is complex and highly variable [37]. Moreover, such morphological variation is not necessarily reflected in differences in genetic composition based on DNA markers [35, 41, 42], thus casting doubt on the validity of the number of gill rakers as the basis for a taxonomic designation [42].

**Route 2: Extinction unverified.** There is insufficient information to support the extinction of the long-spined urchin, the smalltooth sawfish from the Northwest Atlantic and Bermuda, and the largetooth sawfish from the Gulf of California.

During the 1980s, the mass mortality of long-spined urchin populations throughout the Tropical West Atlantic was so severe ($> 93\%$ [43]) that the species was regarded as regionally extinct in the Caribbean [3]. However, more recent evidence shows that the species has always been in the region, albeit at very low numbers [44], and that it is currently recovering [45, 46].

The smalltooth sawfish was declared regionally extinct off the U.S. Atlantic West Coast [3, 47]. Nevertheless, this claim is questionable based on: **(1)** recently published records of the species in the Gulf of Mexico (E3 criterion [48, 49]) and Florida (e.g., [50]); **(2)** a scarcity of documented records from Georgia to New York, disputing the residence of the species in that region (D3 criterion); and **(3)** calculated extinction dates indicating that it has the potential to be present until 2061 (E1 criterion [51]). In addition, the extirpation of *P. pristis* (*P. ristis perotteti* was reported in error) from the Gulf of California and Bermuda was first reported in 2000 [47] and later in 2003 [3]. However, the residence of *P. pristis* in these regions has never been supported by any physical or direct evidence [51].

**Route 3: Possibly extinct.** There is evidence of possible disappearance, but not enough to confirm it—the Guadalupe Storm-Petrel.

The Guadalupe Storm-Petrel was recorded on Guadalupe Island, Mexico, during the late 19th and early 20th centuries until it was last seen in 1912 [52]. The hypothesis of its global extinction is supported by **(1)** it not having been sighted despite surveys carried out over several years; **(2)** potential threats posed by the introduction of cats, goats, and feral dogs; and **(3)** its restricted geographical range [53, 54]. However, the species was categorized by the ISAME as possibly extinct because **(1)** sampling efforts have not been exhaustive during the breeding season when there are greater chances of seeing the species [54]; **(2)** the species' nesting behavior (burrows and colonies active at night) makes its detection even more difficult [55]; and **(3)** several seabirds declared extinct have been later rediscovered (e.g., *Fregetta maoriana*, Oceanitidae, and *Pterodroma cahow*, Procellariidae [1, 55]). This is consistent with the Guadalupe

Storm Petrel being categorized as possibly extinct in 2018 on the basis of updated methods for inferring extinctions [56].

**Route 4: Possibly extinct.** Reasonable doubt regarding the species' records (Stage IV-B) —the lost shark.

The lost shark, distributed in the western Central Pacific Ocean, was declared globally extinct [31] based on the absence of records for more than 80 years in Borneo, Thailand, and Vietnam. Nevertheless, there is reasonable doubt suggesting taxonomic uncertainty: **(1)** the species was described on the basis of a few embryos and juveniles (adult stage not represented); **(2)** in cases that are difficult to identify, such as *C. obsolerus*, genetic analyses become necessary as complementary evidence (which are not currently available); and **(3)** no adult individuals were recorded in more than 80 years, despite the species being distributed in highly fished areas (i.e., intensive opportunistic sampling). Something similar has occurred in other well-documented cases, such as *Anampses viridis* and species mentioned in relation to Route 2, where extinction declarations have been questioned on the basis of taxonomic issues [6]. Furthermore, it has been reported crucial identification problems across the family *Carcharhinidae* (e.g., *Carcharhinus coatesi*, *Carcharhinus tjutjot*, and *Carcharhinus cerdale* [31]).

**Route 5. Extinct.** Sufficient evidence supporting an extinction—the largetooth sawfish in U.S. waters and Steller's sea cow.

This classification route implies that the declaration has positively fulfilled all the system's stages. In 2014, the status and potential for extinction of largetooth sawfish throughout the Atlantic (including its regional extinction in the U.S. waters) was addressed [57]. In the latter region, the species was last seen in 1961, and the estimated more recent upper limits of the extinction date (applying five different statistical methods) do not exceed the current year. Likewise, numerous records (41 documented sightings in Texas, Louisiana and Florida) strongly suggest that the species does reside in the region (D3 criterion). Its hypothesized regional extinction is supported by the fact that, despite being an easily detectable species (large body size and toothed rostrum that tends to become entangled in fishing nets), it has not been sighted for more than 50 years, and the available statistical estimates of the extinction dates indicate that the possibility of sighting the species again is almost zero [57].

The case of Steller's sea cow is an undisputable modern marine extinction caused by overexploitation. The species was discovered in 1741 and was hunted to extinction in less than 30 years [58]. The species was categorized as extinct by ISAME because **(1)** it has not been seen for more than 250 years; **(2)** it is easily detectable and has a relatively restricted geographical range; and **(3)** the upper limit of its extinction date was 1804 [59], far preceding the current year.

## Discussion

We compared the categories assigned to each of our 10 case-study species assessed by ISAME with those from other assessments ([1, 3, 33]—Table 3). We found that 5 out of the 6 species declared extinct by [3] were categorized as unverified by ISAME—and that only the extinction of Steller's sea cow was consistent between the two assessments. This discrepancy was to be expected given that some of the evaluation criteria used by ISAME were not considered by [3] or had not yet been published when they carried out their assessments (e.g., emigration, changes in the distribution, and dates of extinction). In contrast, the extinction assessments by [1] coincided with the categories assigned by ISAME because they were based on a broader and updated review of the available evidence. However, the assessments performed by [1] lacked the comprehensive step-by-step instructional methodology and the additional category of possibly extinct that is inherent in the ISAME assessments.

**Table 3. Comparison of the categories[†] of the ten cases assessed by the integrated system for assessing marine extinctions (ISAME) with respect to other assessments (A [3], B [1], C [33]).**

| Species (extinction level)[‡] | A [3] | B [1] | C [33] | ISAME |
|---|---|---|---|---|
| *Coregonus oxyrinchus* houting (G) | - | - | E | U |
| *Littoraria flammea* periwinkle (G) | E | E | E | U |
| *Diadema antillarum* long-spined urchin (L) | E | U | - | U |
| *Pristis pectinata* smalltooth sawfish (L) | E | U | - | U |
| *Pristis pectinata* smalltooth sawfish (L) | E | U | - | U |
| *Pristis pristis* largetooth sawfish (L) | E | U | - | U |
| *Oceanodroma macrodactyla* Guadalupe storm-petrel (G) | - | - | CR (PE) | PE |
| *Carcharhinus obsolerus* lost shark (G) | - | - | CR (PE) | PE |
| *Pristis pristis* largetooth sawfish (L) | - | - | - | E |
| *Hydrodamalis gigas* Steller's sea cow (G) | E | E | E | E |

[†]Categories: unverified (U), critically endangered (CR), possibly extinct (PE), extinct (E)

[‡]Extinction level: L (local), G (global)

Of the 10 species we assessed, 4 were also assessed by the IUCN. Our results concur with those of the IUCN for the Guadalupe Storm-Petrel (possibly extinct) and Steller's sea cow (extinct). However, we differ with the IUCN listing of the houting and periwinkle as extinct. Running the available data through ISAME indicates that the status of these two species should be unverified extinctions. The difference in the two assessments may be due to the IUCN not considering more recent information—or not having used their new recently-implemented quantitative, systematic methodology (which has only been applied to a few marine species [2]). We suggest that the IUCN consider reassessing the periwinkle and houting.

As with the periwinkle and houting, we also consider three purported extinctions to be questionable (largetooth sawfish in the Gulf of California and smalltooth sawfish in the North-west Atlantic and Bermuda) due to factors associated with migration and the dynamic distributions of the species (D3 criterion). Similar cases to these were previously addressed by [1]. The IUCN defines extinction "when there is no reasonable doubt that the last individual has died". However, the IUCN makes no mention of considering this condition in the case of local extinctions. Some organisms may disappear at a certain site, which can be interpreted as a local extinction, when in fact the disappearance is only due to the mobility of the species within its natural range. The definitions of extinction we propose (Table 1) could complement the understanding of extinctions at local spatial scales.

The method we used to assign numerical values to the criteria resulted in an interesting by-product—namely that these values are the relative weights of each established criteria for defining a species as extinct or otherwise. They show the most significant pieces of information needed to classify a species as extinct are the intensity of the sampling effort carried out to locate a species, and the availability of estimated dates of extinction (A1 and A2 criteria). This was evident for the largetooth sawfish in U.S. waters and the Steller's sea cow. The weightings also show the factors that carry significant importance in questioning an extinction are the presence of sighting records and attributes related to the species' dynamic geographical distribution (D criteria). This was illustrated by the declared extinctions of the long-spined urchin, the smalltooth sawfish in Bermuda and Gulf of California and the largetooth sawfish in North-west Atlantic.

The features that carry the most significant relative weights within ISAME's criteria align with the attributes used by expert assessors to declare the extinction of species across a diversity of taxa [60]. These attributes include the presence of the species given certain search effort,

time from last sighting, and easiness to find in the field (Fig 2), as well as the physical space where the species is and can be found (i.e., changes in the species geographical range), and population decline. Had ISAME been used to analyze the case of the near-extinction of the barndoor skate (*Dipturus laevis*) in 1998 [61], it would not have made the headlines it did because criteria D2 and D3 were not met. The ISAME criteria do not include demographic trends over time, because extinction is viewed as a binary state, meaning that even if a population drastically declines, the species is still considered extant.

## The integrated system to assess marine extinctions

ISAME is a practical and easy-to-use method to synthesize scientific information needed to determine whether a species suspected of being extinct is actually "extinct", "possibly extinct" or "unverified". In contrast, the IUCN's Red List guidelines assign species a category along a gradient that represents different levels of risk (least concern, near threatened, vulnerable, endangered, extinct, etc.). Moreover, some authors have adapted the IUCN guidelines by incorporating local circumstances to direct conservation policies that properly reflect local risk levels because Red List categories do not always coincide with the country-level statuses of species [62, 63]. Thus, while the ISAME and species risk assessments both include the category "extinct", the ISAME method provides a measure of confidence (i.e., whether extinction is unverified, likely, or definite), and complements current assessment methods used by the IUCN and others.

Lists of threatened, endangered and extinct species are used to guide resource allocations for conservation and management [64], and the IUCN Red List, in particular, has helped assess the environmental risks of project developments and ultimately economic investments [65]. However, concerns have been expressed that the benefits that such lists bring to conservation and management might be undermined if extinction assessments are revised and improved [66]. Such concerns have not stopped the IUCN from making several subsequent amendments and corrections to their guidelines (e.g. [16]). Nor should they stop considering alternative approaches for assessing marine extinctions that complement existing ones, and use the most diverse and robust scientific information possible to support conservation policies of socioeconomic relevance.

The ISAME method we propose is not a rebuttal to Dulvy's outstanding efforts to quantify the loss of marine biodiversity at the population and species level [3, 32], but is a complementary assessment method that incorporates additional criteria and data sources. We did not assess any non-marine declarations of extinction, and primarily developed our criteria from information associated with 160 presumed extinctions of species that evolved in the marine realm (see the rationale for criteria in S1 Table). However, some of the criteria we included were also supported by general information on extinction-related topics and reflect methods applicable to marine and nonmarine species alike. For example, criteria A2a or E1 invoke the results of statistical methods that are equally applicable to assessing marine and terrestrial extinctions (e.g., [9, 14, 26]). Likewise, criteria B1, C1 and C2 are easily generalized [28]. As such, ISAME can be confidently used to evaluate marine populations or species—and may be equally applicable to terrestrial species.

We recognize that ISAME has at least three limitations. First, it is possible that a species being assessed may be one considered "contentious". Such cases classified using the ISAME approach (<1%, or 20 of the 2,028 possible criteria combinations) should be done with caution and substantiated with comprehensive evidence. A second limitation is the assignment of numerical values to the criteria to define dichotomic thresholds for Stage IV assessments. While we believe the values we chose are reasonable and valid, we also recognize that future assessments may refine the weighted-importance of each criterion and the interdependence

between them. Lastly, as much as we tried to avoid idiosyncrasies, we acknowledge that some criteria may still be subject to personal interpretation. For example, a constant sampling effort could well be one survey every 5 years over 10 years, or one every year—or the decision between possibly extinct and extinct may switch depending on what is considered reasonable doubt.

Overall, we consider the limitations of ISAME to be minor in light of the strength of clarity that comes from using the ISAME approach—and the simple set of binary responses and combinations of integer values (relative to thresholds) it uses to rapidly assess the likelihood that species are extinct.

## Conservation implications

Some authors argue that extinctions will accumulate [67, 68] to the point of becoming a larger environmental issue than climate change [69]. However, accurate estimates of extinction hinge on having reliable means of assessing whether a population or species has disappeared. They contribute to measuring the effectiveness of conservation efforts [70, 71], and aid in allocating resources for biodiversity research [14, 71]. For example, unverified extinctions highlight information gaps [6, 35] and the need for more sampling [2, 29]. Identifying species that are possibly extinct also helps to reduce the possibility of making the Romeo error (i.e. categorizing a species as extinct when in fact it is not, and therefore not applying conservation strategies to prevent its extinction [72]), while also revealing where the most effective conservation efforts might be directed [8]. Ultimately, assessment methods that use all available evidence may ultimately prove to be an effective means to uncover the key processes and threats associated with extinctions, as well as what ought to be done to prevent future losses of biodiversity.

Confidence in declarations of extinction can be increased if the results of multiple assessment methods are compared to identify similarities and discrepancies—and the relative weights and quality of information incorporated into the different assessments are compared and better understood. Such an approach can also provide greater insights into the strengths and weaknesses of the different approaches in terms of their effectiveness, timeliness and applicability. There appears to be no shortage of opportunity to undertake such tasks. There are currently 150–700 cases of suspected local extinctions that need to be thoroughly assessed using the best documented methods [4, 5].

## Conclusions

We developed a scoring system to assess whether marine species thought to be extinct are actually extinct, possibly extinct, or unverified. Our system consists of a series of binary questions and a numerical threshold nested in different levels or stages, based on quantitative and qualitative scientific information and various technical aspects of the species concerned. Each question and threshold are assigned numerical values and the scoring combination which were determined through an exhaustive computer search using combinatorial methods over a range of integers from which the simplest combinations were chosen i.e., the smallest magnitudes. This searching procedure also indicated that sampling effort and estimated dates of extinction are crucial for declaring a species extinct, while the existence of sighting records and information on the species' distribution dynamics carry more weight to cast doubt on an extinction.

Compared to other existing methods, our ISAME method performs well, and was consistent in 8 out of 11 categorizations made by [1, 33]. ISAME is rigorous and easy to use, and requires less data than other methods (which makes it time saving). Finally, ISAME is a valuable and simple first approach to identify key threats to marine biodiversity and ways to prevent them, and can be readily combined with other extinction assessment methods to thoroughly investigate and assess claims of marine extinctions.

## Supporting information

**S1 Table. Rationale of the ISAME.**
(DOCX)

**S2 Table. Contentious cases.**
(DOCX)

**S1 File. All criteria configurations.**
(XLSX)

**S2 File. Assessing cases.**
(XLSX)

**S3 File. ISAME application.**
(XLSX)

## Acknowledgments

FGM, CHC and PdML thank scholarships of EDI and COFAA. Thanks to José de La Cruz and Roberto Carmona for their comments on the manuscript.

## Author Contributions

**Conceptualization:** Arturo Yáñez-Arenas, Miguel Nakamura, Andrew W. Trites, Pablo del Monte-Luna.

**Data curation:** Arturo Yáñez-Arenas, Héctor Reyes-Bonilla, Claudia Janetl Hernández-Camacho, Felipe Galván-Magaña, Jost Borcherding.

**Formal analysis:** Arturo Yáñez-Arenas, Miguel Nakamura, Pablo del Monte-Luna.

**Funding acquisition:** Miguel Nakamura, Pablo del Monte-Luna.

**Investigation:** Arturo Yáñez-Arenas, Miguel Nakamura, Andrew W. Trites, Héctor Reyes-Bonilla, Felipe Galván-Magaña, Jost Borcherding, Pablo del Monte-Luna.

**Methodology:** Arturo Yáñez-Arenas, Miguel Nakamura, Claudia Janetl Hernández-Camacho, Jost Borcherding, Pablo del Monte-Luna.

**Resources:** Pablo del Monte-Luna.

**Visualization:** Arturo Yáñez-Arenas.

**Writing – original draft:** Arturo Yáñez-Arenas, Miguel Nakamura, Andrew W. Trites, Héctor Reyes-Bonilla, Claudia Janetl Hernández-Camacho, Felipe Galván-Magaña, Jost Borcherding, Pablo del Monte-Luna.

**Writing – review & editing:** Arturo Yáñez-Arenas, Miguel Nakamura, Andrew W. Trites, Claudia Janetl Hernández-Camacho, Jost Borcherding, Pablo del Monte-Luna.

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
