## [Decision Letter · Decision Letter 0]

8 Aug 2023

PONE-D-23-16667AN INTEGRATED SYSTEM TO ASSESS MARINE EXTINCTIONSPLOS ONE

Dear Dr. Del Monte-Luna,

Thank you for submitting your manuscript to PLOS ONE. After careful consideration, we feel that it has merit but does not fully meet PLOS ONE’s publication criteria as it currently stands. Therefore, we invite you to submit a revised version of the manuscript that addresses the points raised during the review process.

The reviewers provide helpful comments to improve the manuscript. Please address all of the reviewers' suggestions, or justify why they cannot be incorporated. A number of Reviewer 2's suggestions are important for readers to be able to understand the methodology better. 

We look forward to receiving your revised manuscript.

Kind regards,

John A. B. Claydon, Ph.D.

Academic Editor

PLOS ONE

Journal Requirements:

3. We note that Figure 2 in your submission contain copyrighted images. All PLOS content is published under the Creative Commons Attribution License (CC BY 4.0), which means that the manuscript, images, and Supporting Information files will be freely available online, and any third party is permitted to access, download, copy, distribute, and use these materials in any way, even commercially, with proper attribution. For more information, see our copyright guidelines: http://journals.plos.org/plosone/s/licenses-and-copyright.

Reviewers' comments:

Reviewer's Responses to Questions

**Comments to the Author**

1. Is the manuscript technically sound, and do the data support the conclusions?

Reviewer #1: Yes

Reviewer #2: Yes

2. Has the statistical analysis been performed appropriately and rigorously? 

Reviewer #1: I Don't Know

Reviewer #2: I Don't Know

3. Have the authors made all data underlying the findings in their manuscript fully available?

Reviewer #1: Yes

Reviewer #2: Yes

4. Is the manuscript presented in an intelligible fashion and written in standard English?

Reviewer #1: Yes

Reviewer #2: Yes

5. Review Comments to the Author

Reviewer #1: The provided papers reports the classification and assessment of presumed extinct species using the ISAME (International Standard for the Assessment of Marine Extinction) method. The authors compared the extinction categories assigned to 10 case-study species by ISAME with those from other assessments, such as previous studies and the IUCN (International Union for Conservation of Nature).

The text discusses the importance of using a standardized methodology, like ISAME, to assess and categorize extinctions consistently and comprehensively. It also emphasizes the need for continuously updating and improving extinction assessments with the latest available information and methods.

Accurate and comprehensive extinction assessments are fundamental for effective biodiversity conservation. They help prioritize conservation efforts, allocate resources, and guide strategies to safeguard marine ecosystems and the species they support. The development and application of assessment methods like ISAME provide valuable tools for addressing the urgent challenges of biodiversity loss and preserving marine life for future generations. I just missed some future work in order to justify all this exercise that you are reporting in this paper.

Reviewer #2: Summary

Yáñez-Arenas et al. present a novel workflow to evaluate marine extinctions that can be applied to any taxonomic level and to both local and global extinctions. The manuscript fills a crucial gap in conservation science, aiming to render extinction claims to be more consistent and systematic. However, the manuscript in its current form lacks “text flow”, has many repetitions and could majorly benefit from restructuring. While the introduced approach offers many advantages to existing assessments, they are not obvious to the reader and should be highlighted in more detail.

Major notes

Major parts of the introduction are discussing pros and cons of qualitative versus quantitative assessments of marine extinctions (line 45 to 79). The authors did a great job in introducing these concepts. I then expected that the authors briefly describe their novel approach in the next paragraph, classify their workflow into one of these categorize (or in-between), and clearly state advantages. Instead, the next paragraph (line 81 to 90) is about the IUCN Red List and whether it makes sense to update their guidelines, followed by two sentences stating the general aims of the manuscript. Beyond this, the authors do not explain their own approach or how their workflow aims to overcome existing problems. The reader is left to decide what the research gap is that the authors are trying to fill. Later on, the authors are stating exactly that (line 100 to 124 and line 131 to 139) in the methods section but this should be moved to the introduction. I therefore suggest to:

- revise the IUCN paragraph (line 81 to 90) and move it to the discussion section

- extend the concluding paragraph so that it becomes more obvious

o what the proposed workflow consists of (i.e. a series of weighted binary criteria)

o where the difference/ novelty of the proposed workflow is compared to existing qualitative or quantitative assessments (i.e. build a bridge to the previous paragraphs).

o that the proposed workflow aims at providing a measure of confidence (in contrast to a gradient of extinction risk sensu IUCN Red List, which is stated in line 497 to 507) and should be explained much earlier

o what the main deliveries and advantages of the proposed workflow is (i.e. can be applied to local and global level and independent of taxonomic level?)

- add a glossary with the terminology of extinction (i.e. what is meant by extinction, what is extirpation, sampling effort, etc.)

Accordingly, the first part of the Materials and Methods section (line 100 to 142) is what I actually expected to see in the Introduction section and does (in my opinion) not belong to methods. I suggest to move it to the Introduction (see remarks above), and start the Materials and Methods section at line 139.

The scoring process of the individual criteria (line is 195 to 218). This relative weighting is the core part of the proposed workflow but it is unclear how the authors came to the values for , , , , . The authors state that they “… coded an exhaustive computer search by 204 combinatorial methods over a range of integers” and then “… identified the simplest combinations”. No more information (or even better, code to recreate the results) is given. I consider myself an expert on statistics could not understand what the authors did here given the description provided.

The authors state that the relative weighting scores can be interpreted as the relative importance of the criteria (line 213 to 215). Unfortunately, this is never discussed in the remaining manuscript and deserves more attention. This is basically telling what criteria are important when considering a taxon extinct and therefore is at the core of the manuscript but not interpreted or discussed at all.

The section “Case studies” (line 389 to 494) does belong to the results section and not the Discussion section. It is currently interrupting the flow of the discussion section and should therefore be moved.

Minor notes

Line 39: “Quantitative approaches are based on frequentist or Bayesian statistics …”. This is ignoring the growing literature on machine learning methods to assess extinction reliability.

Line 92: The authors consider their approach to be qualitative but it heavily relies on numerical scores of criteria and previous statistical analysis. I would argue that ISAME is neither qualitative nor quantitative but instead uses approaches from both.

Line 107: “Some of this data have not been factored into some previous assessments”. This is very vague and it would be interesting to read what kind of data has been added and in comparison to which assessment.

Line 126 and 252: The headings appear to be the same format as the main sections (Introduction, Methods etc) but are subsections of Materials and Methods. They should therefore be made italic.

Line 550: The sentence mentions the “Romeo error” but does not explain what that is. It would be very helpful to have a short explanation (could be added to the above-mentioned glossary with the terminology of extinction)

Line 568: The authors state that “Compared to other existing methods, our assessment performs similarly well (>80%)”. It is unclear what that 80% means.

6. PLOS authors have the option to publish the peer review history of their article (what does this mean?). If published, this will include your full peer review and any attached files.

Reviewer #1: **Yes: **Teresa Luísa Silva

Reviewer #2: No

---

## [Author Response · Author response to Decision Letter 0]

9 Oct 2023

REVIEWER 1

I just missed some future work in order to justify all this exercise that you are reporting in this paper.

Agree. Now, in lines 600-607 we suggest three areas of scientific investigation for potential future research.

REVIEWER 2

1. The manuscript in its current form lacks “text flow”, has many repetitions and could majorly benefit from restructuring. While the introduced approach offers many advantages to existing assessments, they are not obvious to the reader and should be highlighted in more detail. Major parts of the introduction are discussing pros and cons of qualitative versus quantitative assessments of marine extinctions (line 45 to 79). The authors did a great job in introducing these concepts. I then expected that the authors briefly describe their novel approach in the next paragraph, classify their workflow into one of these categorize (or in-between), and clearly state advantages. Instead, the next paragraph (line 81 to 90) is about the IUCN Red List and whether it makes sense to update their guidelines, followed by two sentences stating the general aims of the manuscript. Beyond this, the authors do not explain their own approach or how their workflow aims to overcome existing problems. The reader is left to decide what the research gap is that the authors are trying to fill.

Thank you for your feedback on the manuscript. We have made the suggested changes to improve readability and understanding of the main message and methodological issues. Having reorganized the text as suggested, we also adjusted the grammar to suit the new order of ideas. Below is our response to each of your comments.

2. Later on, the authors are stating exactly that (line 100 to 124 and line 131 to 139) in the methods section but this should be moved to the introduction

We relocated and reorganized those paragraphs within the introduction section (lines 81-135).

3. Revise the IUCN paragraph (line 81 to 90) and move it to the discussion section.

We moved that paragraph to the “discussion section”, specifically, where we address issues relating to IUCN assessments; lines 548-557.

4. Extend the concluding paragraph so that it becomes more obvious: what the proposed workflow consists of -i.e. a series of weighted binary criteria; where the difference/ novelty of the proposed workflow is compared to existing qualitative or quantitative assessments (i.e. build a bridge to the previous paragraphs); the proposed workflow aims at providing a measure of confidence (in contrast to a gradient of extinction risk sensu IUCN Red List, which is stated in line 497 to 507) and should be explained much earlier; what the main deliveries and advantages of the proposed workflow is (i.e. can be applied to local and global level and independent of taxonomic level?).

As suggested in comments #1 and #4, we provided a summary of the method and its main benefits in both the introduction and the conclusions sections (lines 119-127 and 623-628). Furthermore, we moved some paragraphs from the methodology to the introduction, aiming to enhance the reader's comprehension of the methods and demonstrated how the suggested system addresses present information deficits.

5. Add a glossary with the terminology of extinction (i.e. what is meant by extinction, what is extirpation, sampling effort, etcetera).

In Table 1 we defined the terms global extinction, local extinction and the three categories given by our system (Unverified, possibly extinct and extinct). We added the term extirpation within the definition of local extinction. The concept of sampling effort is explained in A1 criterion (lines 278-232).

6. Accordingly, the first part of the Materials and Methods section (line 100 to 142) is what I actually expected to see in the Introduction section and does (in my opinion) not belong to methods. I suggest to move it to the Introduction (see remarks above), and start the Materials and Methods section at line 139.

We moved and restructured all those paragraphs to the introduction section (lines 81-135).

7. The scoring process of the individual criteria (line is 195 to 218). This relative weighting is the core part of the proposed workflow but it is unclear how the authors came to the values for , , , , . The authors state that they “… coded an exhaustive computer search by 204 combinatorial methods over a range of integers” and then “… identified the simplest combinations”. No more information (or even better, code to recreate the results) is given. I consider myself an expert on statistics could not understand what the authors did here given the description provided.

We reorganize and add information regarding this issue. The motivation for the numerical scoring was moved to the very beginning. Additional text was included to better explain the construction of the index and finding its corresponding weights. We hope this re-writing makes it now clear that the method is merely a brute-force, computational, exhaustive search for weights and a threshold that together render perfect classification among 2028 combinations, rather than conventional statistical data analysis (including classification techniques). 

8. The authors state that the relative weighting scores can be interpreted as the relative importance of the criteria (line 213 to 215). Unfortunately, this is never discussed in the remaining manuscript and deserves more attention. This is basically telling what criteria are important when considering a taxon extinct and therefore is at the core of the manuscript but not interpreted or discussed at all.

Thank you for pointing out this omission. We added a paragraph in the discussion addressing the relative importance of the criteria (lines 514-534).

9. The section “Case studies” (line 389 to 494) does belong to the results section and not the Discussion section. It is currently interrupting the flow of the discussion section and should therefore be moved.

All “Case studies” were relocated in the results section.

10. Line 39: “Quantitative approaches are based on frequentist or Bayesian statistics …”. This is ignoring the growing literature on machine learning methods to assess extinction reliability.

We rephrased that idea and introduced a new paragraph briefly mentioning those newer techniques (lines 57-64).

11. Line 92: The authors consider their approach to be qualitative but it heavily relies on numerical scores of criteria and previous statistical analysis. I would argue that ISAME is neither qualitative nor quantitative but instead uses approaches from both.

Although ISAME uses quantitative information (from other methods: A2 criterion) and a part of the ISAME is based on obtaining numerical scores (Stage 3). In reality, ISAME is qualitative in nature, as the Stage 3 could be done by simply consulting the large and cumbersome table of 2048 combinations between criteria. In the new material introduced beginning on line 189, we have strived to make it clearer that the method is essentially defined by combinations of 11 binary criteria alone, and could be, in fact, implemented simply by looking up classification results in a table having 2048 rows. The numerical scoring device is only a way to circumvent a table lookup. The numbers described convert the large table into a score for determining the classification of a given case, and they are not a product of previous statistical analysis but rather a convenient, summarized representation of the large table.

12. Line 107: “Some of this data have not been factored into some previous assessments”. This is very vague and it would be interesting to read what kind of data has been added and in comparison to which assessment.

In this sentence we added three examples of factors that have not been taken into account in previous assessments. We also addressed all this in the “Case studies” and in the discussion in lines 416-424, 445-453, 504-513. It can be seen how in some Dulvy et al (2003)'s, IUCN, and White et al (2019) assessments did not consider certain factors. On the contrary, here when applying the ISAME we considered them and therefore questioned the extinction.

13. Line 126 and 252: The headings appear to be the same format as the main sections (Introduction, Methods etc) but are subsections of Materials and Methods. They should therefore be made italic.

The headings format was corrected. 

14. Line 550: The sentence mentions the “Romeo error” but does not explain what that is. It would be very helpful to have a short explanation (could be added to the above-mentioned glossary with the terminology of extinction).

We decided to explain the Romeo error in the body of the text (lines 593-597) since Table 1 is a glossary of extinction-related concepts and the system categories.

15. Line 568: The authors state that “Compared to other existing methods, our assessment performs similarly well (>80%)”. It is unclear what that 80% means.

That figure (80%) comes from the ISAME categorization coinciding with 5 out of 6 species of the IUCN’s assessments. We modified the text flow to make it clearer. We changed it to: the concordance between ISAME categorizations with other assessments. In lines 623-628 can be seen.

---

## [Editor Report · Decision Letter 1]

13 Oct 2023

AN INTEGRATED SYSTEM TO ASSESS MARINE EXTINCTIONS

PONE-D-23-16667R1

Dear Dr. Del Monte-Luna,

We’re pleased to inform you that your manuscript has been judged scientifically suitable for publication and will be formally accepted for publication once it meets all outstanding technical requirements.

Kind regards,

John A. B. Claydon, Ph.D.

Academic Editor

PLOS ONE
---

## [Editor Report · Acceptance letter]

17 Oct 2023

PONE-D-23-16667R1 

An integrated system to assess marine extinctions 

Dear Dr. del Monte-Luna:

I'm pleased to inform you that your manuscript has been deemed suitable for publication in PLOS ONE. Congratulations! Your manuscript is now with our production department. 

Kind regards, 

on behalf of

Dr. John A. B. Claydon 

Academic Editor

PLOS ONE